# Six states of *Enterococcus hirae* V-type ATPase reveals non-uniform rotor rotation during turnover

Raymond N. Burton-Smith[1,2], Chihong Song[1,2,3], Hiroshi Ueno[4], Takeshi Murata [5], Ryota Iino [6,7] & Kazuyoshi Murata [1,2,3]✉

The vacuolar-type ATPase from *Enterococcus hirae* (EhV-ATPase) is a thus-far unique adaptation of V-ATPases, as it performs Na$^+$ transport and demonstrates an off-axis rotor assembly. Recent single molecule studies of the isolated V$_1$ domain have indicated that there are subpauses within the three major states of the pseudo three-fold symmetric rotary enzyme. However, there was no structural evidence for these. Herein we activate the EhV-ATPase complex with ATP and identified multiple structures consisting of a total of six states of this complex by using cryo-electron microscopy. The orientations of the rotor complex during turnover, especially in the intermediates, are not as perfectly uniform as expected. The densities in the nucleotide binding pockets in the V$_1$ domain indicate the different catalytic conditions for the six conformations. The off-axis rotor and its' interactions with the stator a-subunit during rotation suggests that this non-uniform rotor rotation is performed through the entire complex.

[1] Exploratory Research Center on Life and Living Systems (ExCELLS), National Institutes of Natural Sciences, 38 Nishigonaka, Myodaiji, Okazaki, Aichi 444-8585, Japan. [2] National Institute for Physiological Sciences, National Institutes of Natural Sciences, 38 Nishigonaka, Myodaiji, Okazaki, Aichi 444-8585, Japan. [3] Department of Physiological Sciences, School of Life Science, The Graduate University for Advanced Studies (SOKENDAI), 38 Nishigonaka, Myodaiji, Okazaki, Aichi 444-8585, Japan. [4] Department of Applied Chemistry, Graduate School of Engineering, The University of Tokyo, 7-3-1 Hongo, Bunkyo-Ku, Tokyo 113-8656, Japan. [5] Department of Chemistry, Graduate School of Science, Chiba University, 1-33 Yayoi-Cho, Inage-Ku, Chiba 263-8522, Japan. [6] Institute for Molecular Science, National Institute for Natural Sciences, 5-1 Higashiyama, Myodaiji, Okazaki, Aichi 444-8787, Japan. [7] Department of Functional Molecular Science, School of Physical Sciences, The Graduate University for Advanced Studies (SOKENDAI), 5-1 Higashiyama, Myodaiji, Okazaki, Aichi 444-8585, Japan. ✉email: kazum@nips.ac.jp

Most rotary ATPases are a structurally similar but broad ability group of enzymes, which carry out a variety of different cellular processes[1]. They play critical roles within both prokaryotic and eukaryotic organisms, acting for ATP (adenosine triphosphate) synthesis[2] and ATP-driven ion transport[1]. Their ubiquity, critical role in life of a cell, and sophisticated mechanism and design have made them popular targets for pharmacological treatments for a range of diseases from cancers[3], osteoporosis[4], and kidney disease[5] to bacterial and fungal infections[6]. They are classified by type; A-, F-, or V-type, which are not necessarily mutually exclusive in role[7]. For example, a single ATPase can carry out ATP synthesis as well as ion translocation by ATP hydrolysis[8]. The vacuolar-type (V-) ATPases are a ubiquitous class of membrane protein complex which utilise ATP hydrolysis to power ion transport across cellular membranes[9], although the *Thermus thermophilus* (Tt) V/A-ATPase is a dual-role ATPase[10], which usually functions as an ATP synthase[11].

Structures of the TtV/A-ATPase and *Saccharomyces cerevisiae* (Sc) V-ATPases and examinations of their subunits revealed common molecular mechanisms across V-ATPases[12]. V-ATPases are formed from two domains: the $V_o$ and $V_1$ domains. In the V-ATPase from *Enterococcus hirae* (EhV-ATPase), the $V_1$ domain in cytoplasm consists of A-, B-, D-, E-, F-, G-, and d-subunits, and the $V_o$ domain bound membrane consists of a- and c-subunits[13]. The cylindrical "c-ring" is located entirely within the membrane and binds the ion to be transported, rotating almost a full turn to bring the bound ions from ingress to egress. The a-subunit possesses a membrane-integrated region that interacts with the c-ring to control ion binding, and a cytosolic region that interacts with the EG "stalk" complexes to support the cytosolic $V_1$ domain. The $V_1$ domain creates a stator formed with the $A_3B_3$ complex connected to the EG stalks and maintains the rotor shaft formed with the D- and F-subunits. The transport of the bound ion across the membrane is performed by the hydrolysis of ATP in the $V_1$ domain, which drives the rotation of a coupling "rotor" at three pauses separated by 120°, depending on the pseudo trimeric A/B dimer $V_1$ ATPase domain. This discontinuous rotation causes the rotation of the d-subunit and membrane-intrinsic c-ring, resulting in three conformational states in the entire complex. Many structures of V-ATPases have been reported demonstrating the three turnover states of each, both as isolated $V_1$ domains[14–16] and as complete complexes[17–22].

The V-ATPases show adaptations across species; the structures of the membrane-intrinsic c-ring and the cytosolic supporting peripheral stalks vary significantly in subunit composition and arrangement[9]. For example, eukaryotic ScV-ATPase exhibits a large diameter c-ring consisting of 10 quadruple-helix subunits (a total of 40 transmembrane helices, although the subunits are not all homogeneous[23]) and three rigid supporting stalks[17], while prokaryotic TtV/A-ATPase exhibits a smaller diameter c-ring consisting of 12 twin-helix subunits (a total of 24 transmembrane helices) and just two supporting stalks[19]. The EhV-ATPase is specific for $Na^+$[24] rather than acting to control $H^+$ gradient across the membrane[25–27]. Subunit composition of the EhV-ATPase has been previously reported[28], being the same as the subunit composition of TtV/A-ATPase except for the number of subunits in the c-ring. The atomic structure of the c-ring membrane-bound domain was elucidated by x-ray crystallography revealing a decameric c-ring consisting of 10 quadruple-helix subunits[29], showing a eukaryotic-type c-ring but prokaryotic-type stalks. Ion selectivity is controlled by amino-acid residues in the c-subunit ion binding pocket[30]. The structures of the rotor shaft[31] and the pseudo-trimeric A/B dimer $V_1$ ATPase domain[14] were determined by x-ray crystallography and different catalytic states

of the $V_1$ domain demonstrated using non-hydrolysable ATP analogues[15].

EhV-ATPase was previously reported to have no intermediate pauses between the 120° rotations of the DF rotor complex[32,33]. However, more recent work has demonstrated that there are indeed intermediate pauses divided by an ~40°/80° split between the 120° major pauses[34]. Like the F-ATPases[35], which demonstrate subpauses between the distinct rotor positions aligned with the $F_1$ domain, these subpauses are also not precisely midway between the major conformations. The F-ATPases show variability in angular subpauses, ranging from an 80°/40° split for thermophilic *Bacillus* PS3[36], a 65°/25°/30° split for human mitochondrial F-ATPase[37], an 85°/35° split for *Escherichia coli*[38], and an 87°/33° split for yeast mitochondrial F-ATPases[39]. Recently, cryo-electron microscopy (cryo-EM) has visualised the subpause structures of the thermophilic *Bacillus* PS3 $F_1$-ATPase[40]. However, there is still no structural evidence for subpauses in V-ATPases.

Previously, we reported the off-axis rotor[41] of the EhV-ATPase using phase contrast cryo-EM single particle analysis (SPA) with a Zernike-type phase plate. This permitted the first visualisation of the detergent-solubilized EhV-ATPase full complex. While this data provided insight into the overall macromolecular organisation of this unusual V-ATPase complex conformation, it was limited by resolution constraints. The off-axis nature of the rotor has thus far only been identified in EhV-ATPase[41]. Although other ATPases can demonstrate both the rotor and c-ring being off-axis for the $V_1$ cytoplasmic domain[42,43], EhV-ATPase is the only one identified with an "on axis" c-ring but off axis rotor. It was suggested to be caused with the asymmetric stalks and the large c-ring against the rotor shaft, but the physiological function was not clear[29,33].

Here, we present six state structures of the EhV-ATPase complex in three major states, corresponding to the three reported conformations in TtV/A-ATPase[19], at resolutions of 4.2 to 4.4 Å, and in newly clarified three intermediate states at resolutions of 4.8 to 7.7 Å. Focussed refinements of the $V_1$ domain improved the resolution to 3.8–4.1 Å in the major states. Importantly, these states were all identified using the same aqueous conditions from the same preparation and grids, while many other works identifying intermediates utilise different aqueous conditions and substrates to promote the different conformations[44]. These results provide innovated insights into the V-ATPases, where the orientations of the rotor complex during turnover, especially in the intermediates, were not as perfectly uniform as expected. The off-axis rotor and its interactions with the stator a-subunit during rotation suggests that this non-uniform rotor rotation is performed through the entire complex.

## Results

**Data collection and 3D reconstruction of cryo-EM SPA for EhV-ATPase.** Figure 1 and inset show the structure of the entire EhV-ATPase domain and its off-axis rotor scheme. To attempt to generate all catalytic states of EhV-ATPase, ATP, $Mg^{2+}$, and $Na^+$ were added to the purified and detergent solubilized complex. Seven mM ATP and 100 mM $Na^+$ were used to detect as many possible EhV-ATPase states during turnover. In single-molecule studies, EhV-ATPase exhibits rotor rotation in the presence of these concentrations of ATP and $Na^+$[33,45]. Our previous work showed that the EhV-ATPase complexes with peptide tag inserted into the rotor D-subunit exhibited different states with and without Fab binding upon activation of the complex with these concentrations of ATP and $Na^+$[41]. The sample was vitrified during the active state transition. A total of 47,372 micrograph

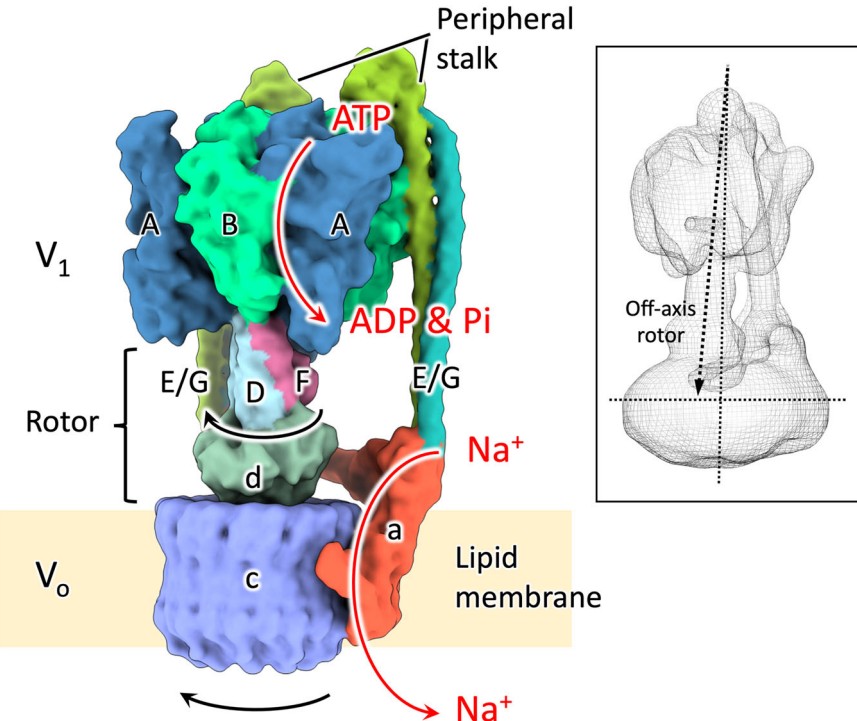

**Fig. 1 Schematic drawing of the EhV-ATPase.** Subunits are individually colour-coded and labelled. A-, B-, D-, E-, F-, G- and d-subunits form the $V_1$ domain, while a- and c-subunits form the $V_o$ domain[13]. The a-subunit and c-ring are embedded in lipid membrane. The rotation of the D/F/d rotor shaft and c-ring proceeds clockwise when viewed from $V_1$ to $V_o$ domains, as indicated. Turnover is driven by the entry of ATP into a binding pocket at the interface of each A/B dimer, the hydrolysis reaction of which drives conformational changes which cause the rotation of the rotor. Inset shows the off-axis rotor previously reported[41].

movies were acquired using SerialEM[46] on a JEOL CRYO ARM 300 electron microscope equipped with a Gatan K3 direct detector in three sessions (Supplementary Table 1). The detergent solubilized EhV-ATPase was a very low contrast complex, which made it difficult to see in conventional defocus-acquired cryo-EM (Supplementary Fig. 1a). As a result, the particle picking strategy used was more liberal, with a lot of undesirable particles in the selection of 4,383,273 particle candidates (Supplementary Fig. 1b, Table 1 and Supplementary Table 1). These particles were cleaned through multiple rounds of two-dimensional (2D) and three-dimensional (3D) classification. The processing strategy covered in detail in "Methods" and Supplementary Fig. 1 broadly follows the standard processing workflow of RELION 3.1 software[47–50]. Interestingly, without the addition of $Na^+$, the purified EhV-ATPase would tear itself apart upon addition of ATP. Despite this, the complex was still very fragile, and a large percentage of particles were removed during classification due to evident damage to either the rotor shaft or the $V_o$ domain (Supplementary Fig. 1, Table 1 and Supplementary Table 1). The ice-embedded complex showed a strong orientation preference (Supplementary Fig. 2a), which limited the resolution and quality of reconstructions with the first acquired dataset. For this reason, the third dataset was acquired with both 20° and 30° stage tilt to attempt to overcome this orientation preference (Supplementary Table 1). It was mostly successful (Supplementary Fig. 2b) and provided sufficient extra variance in views to improve the angular sampling of reconstructions (Supplementary Fig. 2c).

Focussed refinement using a soft mask containing only the $V_1$ region of the complex improved resolution to 3.8–4.1 Å for this region in the major states by excluding the flexible peripheral stalks and $V_o$ (membrane integrated) domain (Supplementary Figs. 3 and 4), although the d-subunit density forming the terminus of the rotor shaft was still poorly resolved

(Supplementary Fig. 3). Attempting to improve the clarity of the $V_o$ domain by repeating this focussed refinement strategy using a mask for the $V_o$ domain failed, even when falling back to a recombined particle set, masking out the $V_1$ domain, and re-centring during initial 3D classification. Similarly, an attempt to better resolve the peripheral stalks by focussed refinement failed to converge, although this is unsurprising given that the $V_1$ and $V_o$ domains present a much more intense signal for alignment than the weak signal of the stalks.

**Examination of six states of EhV-ATPase at F-subunit position.** The major three states of EhV-ATPase isolated were reconstructed to 4.2–4.4 Å (global) resolution (Fig. 2 and Supplementary Figs. 1, 3 and 4) with the final maps consisting of 100,000–260,000 particles (15–37% of all particles) (Table 1). These three states correspond to the rotor orientations of conformations 1, 2 and 3 of TtV/A-ATPase[19], and thus were defined as the major states 1, 2 and 3 with the orientations of the F-subunit (red asterisks in Fig. 2). These three major states were expected to rotate the rotor in a 120° rotation step to correspond to the pseudo three-fold symmetric rotary enzyme, but the cumulative rotation angle of 227° in State 3 was 13° smaller than the expected cumulative rotation of 240° (Fig. 2 and Table 2).

The three intermediate states named State 1', State 2', and State 3', were identified by the orientation of the rotor with the F-subunit between the three major states of the EhV-ATPase (red asterisks in Fig. 2). These intermediate states are each composed of a similarly small number of particles (~50,000, Table 1). These states may be considered transitory because of the lower particle count (<7.4%) and the lower resolution (>4.8 Å) compared to the major three states (Fig. 2 and Supplementary Figs. 3 and 4). Recent single-molecule imaging studies of $V_1$ domain indicated that there are subpauses divided

**Table 1 Cryo-EM data collection and validation statistics.**

| Data collection and processing | #1 EhV-ATPase state 1 (whole complex) (EMDB-34139) | #2 EhV-ATPase state 1' (whole complex) (EMDB-34140) | #3 EhV-ATPase state 2 (whole complex) (EMDB-34141) | #4 EhV-ATPase state 2' (whole complex) (EMDB-34142) | #5 EhV-ATPase state 3 (whole complex) (EMDB-34143) | #6 EhV-ATPase state 3' (whole complex) (EMDB-34144) | #7 EhV-ATPase state 1 ($V_1$ domain) (EMDB-34145) | #8 EhV-ATPase state 1' ($V_1$ domain) (EMDB-34146) | #9 EhV-ATPase state 2 ($V_1$ domain) (EMDB-34147) | #10 EhV-ATPase state 2' ($V_1$ domain) (EMDB-34148) | #11 EhV-ATPase state 3 ($V_1$ domain) (EMDB-34149) | #12 EhV-ATPase state 3' ($V_1$ domain) (EMDB-34150) |
|---|---|---|---|---|---|---|---|---|---|---|---|---|
| Magnification | 50,000 | 50,000 | 50,000 | 50,000 | 50,000 | 50,000 | 50,000 | 50,000 | 50,000 | 50,000 | 50,000 | 50,000 |
| Voltage (kV) | 300 | 300 | 300 | 300 | 300 | 300 | 300 | 300 | 300 | 300 | 300 | 300 |
| Electron exposure (e$^-$/Å$^2$) | 50–60 | 50–60 | 50–60 | 50–60 | 50–60 | 50–60 | 50–60 | 50–60 | 50–60 | 50–60 | 50–60 | 50–60 |
| Defocus range (µm) | −1–−2 | −1–−2 | −1–−2 | −1–−2 | −1–−2 | −1–−2 | −1–−2 | −1–−2 | −1–−2 | −1–−2 | −1–−2 | −1–−2 |
| Pixel size (Å) | 1.01 | 1.01 | 1.01 | 1.01 | 1.01 | 1.01 | 1.01 | 1.01 | 1.01 | 1.01 | 1.01 | 1.01 |
| Symmetry imposed | C1 | C1 | C1 | C1 | C1 | C1 | C1 | C1 | C1 | C1 | C1 | C1 |
| Initial particle images (no.) | 4,383,273 | 4,383,273 | 4,383,273 | 4,383,273 | 4,383,273 | 4,383,273 | 4,383,273 | 4,383,273 | 4,383,273 | 4,383,273 | 4,383,273 | 4,383,273 |
| Final particle images (no.) | 185,127 | 46,901 | 257,397 | 47,515 | 110,166 | 51,459 | 185,127 | 46,901 | 257,397 | 47,515 | 110,166 | 51,459 |
| Map resolution (Å)/FSC threshold | 4.3/0.143 | 7.7/0.143 | 4.3/0.143 | 4.8/0.143 | 4.4/0.143 | 7.3/0.143 | 3.8/0.143 | 7.7/0.143 | 3.9/0.143 | 4.2/0.143 | 4.1/0.143 | 7.0/0.143 |
| Map resolution range (Å) | 4.16–5.61 | 7.59–8.42 | 4.30–5.72 | 4.76–6.78 | 4.40–6.54 | 7.15–8.06 | 3.80–4.34 | 7.58–8.12 | 3.87–4.38 | 4.18–6.30 | 4.07–5.56 | 6.97–7.69 |

by an ~40°/80° split within the major 120° rotation pauses of the pseudo three-fold symmetric rotation enzyme[34]. The orientations of the rotor in the major states were close to these values in the 120° step, while the orientations of the rotor in the intermediate states were diverse (Fig. 2 and Table 2). Compared to the expected +40° subpause from the previous major step, the 50°, 191°, and 307° rotor orientations at the F-subunit of States 1', 2', and 3' showed subpause shifts of +10°, +31°, and +27°, respectively (Table 2).

**Examination of six states of EhV-ATPase at $V_1$ position.** The orientations of the rotor D-subunit based upon the direction of the two long helices showed similar angles as the position of the rotor F-subunit in both the major and intermediate states in the $V_1$ domain (red boxes in Fig. 3 and Table 2). The result shows the rotor D-subunit is not elastic but rigid between the $V_1$ domain and the F-subunit (Supplementary Fig. 5). The cryo-EM maps at the resolution from 7.3 to 3.8 Å clearly illustrated the conformations of the A/B subunit in $V_1$ domain in the pseudo three-fold symmetric rotary enzyme (Fig. 3 and Table 1). These are classified as "Open", "Closed", and "Semi-closed" based on the previous report of the crystallographic models of the $V_1$ domain[15], each showing the different ATPase catalytic condition. In the major three states, each A/B subunit takes a different conformation, which is switched to rotate in the order of "Open", "Semi-closed", and "Closed", in a continuous cycle between the three states, rotating the central rotor (Fig. 3). Interestingly, in the intermediate $V_1$ states, State 1' and State 3' took the similar conformation with State 1 and State 3, respectively, while State 2' showed a similar conformation to State 3.

**Examination of six states of EhV-ATPase at $V_o$ position and their off-axis assembly.** The orientation angles of the rotor shaft terminus in the six states of EhV-ATPase were estimated with the d-subunit located between D-subunit and c-ring (Fig. 1), which also showed angles like the rotor F-subunit position at the $V_o$ position in both major and intermediate states (Fig. 4 and Table 2). The result indicated the rotor D-subunit and the following rotor shaft terminus d-subunit are not elastic but rigid between the $V_1$ domain and the $V_o$ domain. However, the d-subunit was able to move relative to the c-ring (Fig. 4 and Supplementary Fig. 6), causing off-axis rotation of the rotor. The off-axis nature of the rotor varied during turnover of the EhV-ATPase complex (Fig. 4 and Supplementary Fig. 6). The degree to which the rotor is "off-axis" changed continuously as rotation proceeded. For State 2', the off-axis nature of the rotor was clearly detected, while for State 3' the volume centre of the d-subunit is closer to the centre of the c-ring (red and black crosses in Fig. 4 and Supplementary Fig. 6 and Table 2). Measuring the lateral distance from the centre axis of the c-ring to the volume centre of the d-subunit on the lipid membrane plane (Supplementary Fig. 6), the larger off-axis centres of d-subunit were detected as 8 Å in State 2', while the smaller off-axis centres of the d-subunit were detected as 3 Å in State 3' (Fig. 4, Supplementary Fig. 6 and Table 2). The distance gradually increases from State 3' to State 2', and then decreases rapidly between State 2' and State 3'. This off-axis of the d-subunit can be caused by the interaction between the large c-ring and the d-subunit (red asterisks in Fig. 4), although the interaction residues between these subunits were not clear at the limited resolution of the $V_o$ domain. On the other hand, this off-axis re-centring of the d-subunit can be partially interfered by the extrinsic domain of the a-subunit connected to the peripheral stalks to support the $V_1$ domain (Fig. 5). Interference between the d-subunit of the rotor and the extrinsic domain of

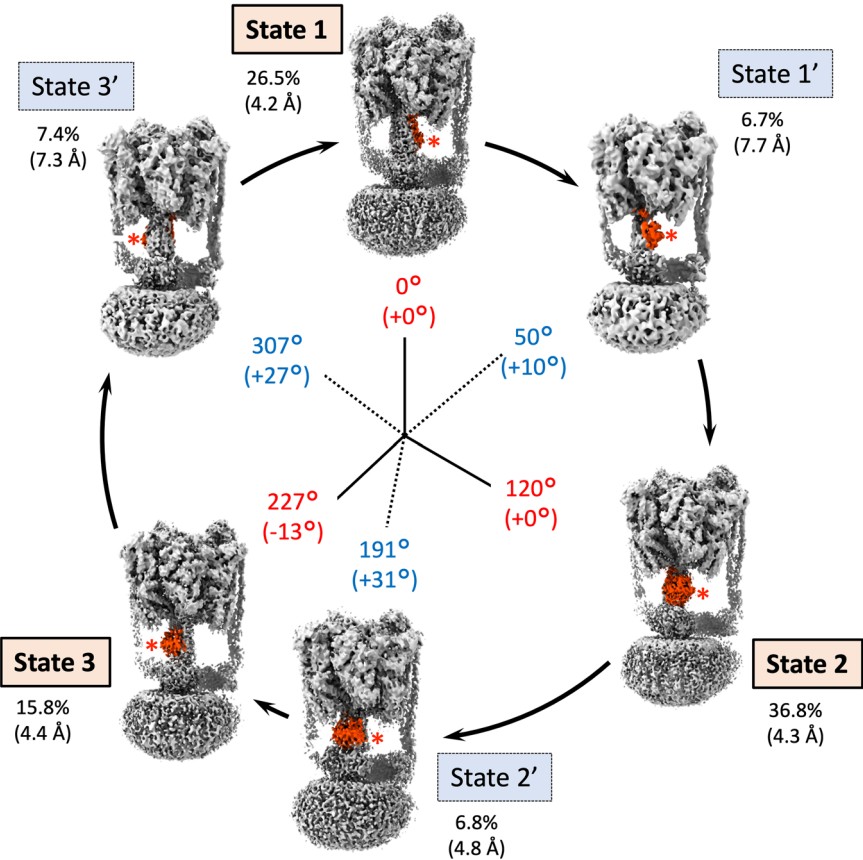

**Fig. 2 The six state structures of EhV-ATPase isolated in this study.** The F-subunit position is highlighted in red for easier identification of orientation of the rotor. Starting in State 1 at "12 o'clock" on the circle and proceeding clockwise when turnover is viewed from the $V_1$ to $V_o$ domains. The six state structures are defined as State 1, State 1', State 2, State 2', State 3, and State 3' with comparisons to the other V-ATPases. Total rotation of the rotor at F subunit is labelled in red for the major states and blue for the intermediate states internally of the circle. The gaps from the orientations based on the single-molecule imaging studies (120° major pauses, and 40/80° subpauses in the major pauses) are in brackets. Relative percentages of the total final particles used and their resolutions (brackets) for each reconstruction are indicated externally of the circle. The cryo-EM maps of EhV-ATPase are aligned according to the orientation of the F-subunit.

**Table 2 Sumary of non-uniform rotor rotation.**

|  | State 1 | State 1' | State 2 | State 2' | State 3 | State 3' |
|---|---|---|---|---|---|---|
| Orientation angle of the D-subunit in $V_1$ domain | 0° (0°) | 55° (+15°) | 120° (0°) | 194° (+34°) | 234° (−6°) | 309° (+29°) |
| Orientation angle of the F-subunit in rotor | 0° (0°) | 50° (+10°) | 120° (0°) | 191° (+31°) | 227° (−13°) | 307° (+27°) |
| Orientation angle of the d-subunit in $V_o$ domain | 0° (0°) | 53° (+13°) | 120° (0°) | 191° (+31°) | 231° (−9°) | 309° (+29°) |
| Average of the above and SD | 0 ± 0° (0°) | 53 ± 3° (+13°) | 120 ± 0° (0°) | 192 ± 2° (+32°) | 230 ± 4° (−10°) | 308 ± 1° (+28°) |
| Distance between the centres of c-ring and d-subunit | 6 Å | 5 Å | 5 Å | 8 Å | 5 Å | 3 Å |
| ATP pocket 1 | Empty | Full | Full | Full | Full | Full |
| ATP pocket 2 | Full | Full | Full | Empty | Empty | Full |
| ATP pocket 3 | Full | Full | Empty | Full | Full | Full |
| (a) E139a to R573 distance (Cα to Cα) | 11.9 Å | 11.4 Å | 12.3 Å | 11.0 Å | 11.8 Å | 10.6 Å |
| (b) E139b to R573 distance (Cα to Cα) | 9.6 Å | 10.5 Å | 9.6 Å | 10.8 Å | 9.8 Å | 11.6 Å |
| Gap (a − b) | +2.3 Å | +0.9 Å | +2.7 Å | +0.2 Å | +2.0 Å | −1.0 Å |

Orientations of the subunit at different positions in the complex during turnover (gap from 120° major pauses or 40/80° subpauses reported by single molecular analysis), distances between the volume centres of d-subunit and c-ring on the lipid membrane plane, presence (or lack) of density in each A/B dimer nucleotide binding pocket, and distances from R573 in a-subunit to E139a or E139b in c-ring and their gaps.

the a-subunit were observed in State 2' and State 3 (blue asterisks in Fig. 5), where the large off-axis nature of the d-subunit (8 Å off-centre) in State 2' was rapidly re-centred (3 Å off-centre) in State 3' (Fig. 4 and Supplementary Fig. 6).

**Examination of six states of EhV-ATPase at the nucleotide binding pocket.** Significant studies have been focussed on the turnover of the $V_1$ domain for EhV-ATPase, permitting the development of models of how turnover proceeds

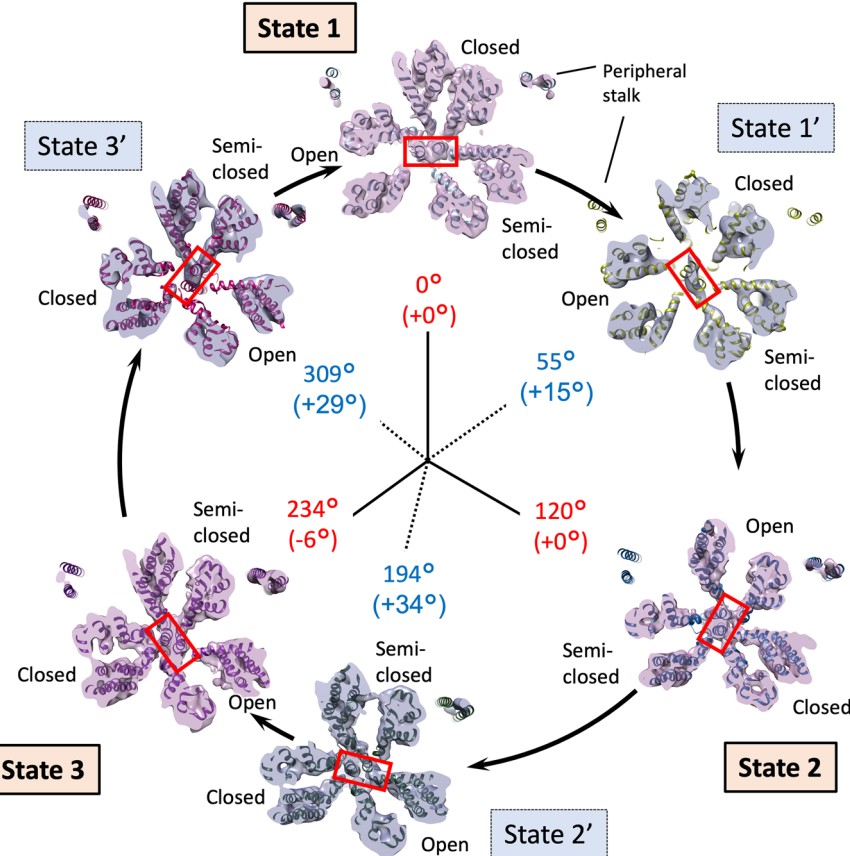

**Fig. 3 V₁ domain cross-section view of the six states.** The figure is laid out as shown in Fig. 2 viewed from the V₁ to Vₒ domains. The rotor D subunit is boxed in red, demonstrating the positions of a pair of the longest helices. The catalytic conformations of the A/B subunit in V₁ domain are indicated with "Open", "Closed", and "Semi-closed". The positions of peripheral stalk are labelled. Cumulative rotation angles of the rotor D subunit at the V₁ domain are labelled as the same as Fig. 2.

mechanistically[33,34,51] and structurally[14,15] during ATP hydrolysis. In Fig. 6, the crystallographic model of the three ADP-bound V₁ complex (PDBID: 5KNC)[15] was fitted into the cryo-EM maps to see whether density for a bound nucleotide is present or not. These images are magnified in Supplementary Figs. 7–12, respectively. A cross-section of the maps across the three nucleotide binding pockets in the V₁ domain directly shows that in the major three states, two nucleotide binding pockets contain densities for nucleotides (blue circles in States 1, 2, and 3 in Fig. 6), but the third one is empty (red circles in States 1, 2, and 3 in Fig. 6). In addition, these three states represented continuous changes in binding/catalytic dwell during turnover. These images are magnified in Supplementary Figs. 7, 9 and 11, respectively, where the existence of nucleotide and its binding pocket formed with the walker A motif[52] was clearly visible in each state (Insets in Supplementary Figs. 7, 9 and 11). In the intermediate states, despite the relatively low resolutions of the V₁ domain (4.2–7.7 Å), we sought to find the nucleotide density in the nucleotide binding pocket. States 1' and 3' appear to contain the densities in all three nucleotide binding pockets, but are less clear at >7 Å resolution (blue dotted circles in State 1' and 3' in Fig. 6, Supplementary Figs. 8 (Inset) and 12 and Table 2), while State 2' at 4.2 Å resolution clearly shows bound coordinates similar to State 3 (State 2' in Fig. 6, Supplementary Fig. 10 and Table 2). This result is consistent with the notion that the major three states correspond to the main pauses waiting for ATP binding, observed in the previous single-molecule study of isolated V₁ domain[34].

## Discussion

Here we identified six independent conformations of EhV-ATPase by cryo-EM using an active sample which was encouraged to turnover through the addition of ATP, Mg²⁺ and Na⁺. While the purified EhV-ATPase complex is fragile, and the actively rotating complex even more so, but the identification of three major states and three intermediate states provides some further structural clues to the activity of this so far unique "off-axis" adaptation to a ubiquitous membrane protein complex.

Single-molecule imaging studies of the isolated V₁ domain of EhV-ATPase have indicated that there are three major pauses of each 120° rotation of the rotor of the pseudo three-fold symmetric rotary enzyme and three intermediate subpauses, each divided by ~40°/80° split within the major pauses[34]. We directly examined these rotor angles of the major three states and the intermediate three states in the six structures. The results show the major pauses for each 120° rotation in State 1 and State 2, but the cumulative angle of ~230° in State 3 was 10° smaller than the expected cumulative angle of 240° at all rotor positions (Figs. 2–4 and Table 2). The orientation of the subpauses was more diverse in the intermediate states. The smallest gap from the expected subpause angle of 40/80° in the 120° rotation was shown in State 1', where the subpause angle of ~53° showed a positive gap of 13° at all rotor positions (Table 2). In contrast, the subpause angles of ~192° and 308° in State 2' and State 3' showed positive gaps of 32° and 28°, respectively, and were significantly larger than that of State 1' (Table 2). These are suggested to be caused by the interference between the off-axis rotor and the stator subunits in

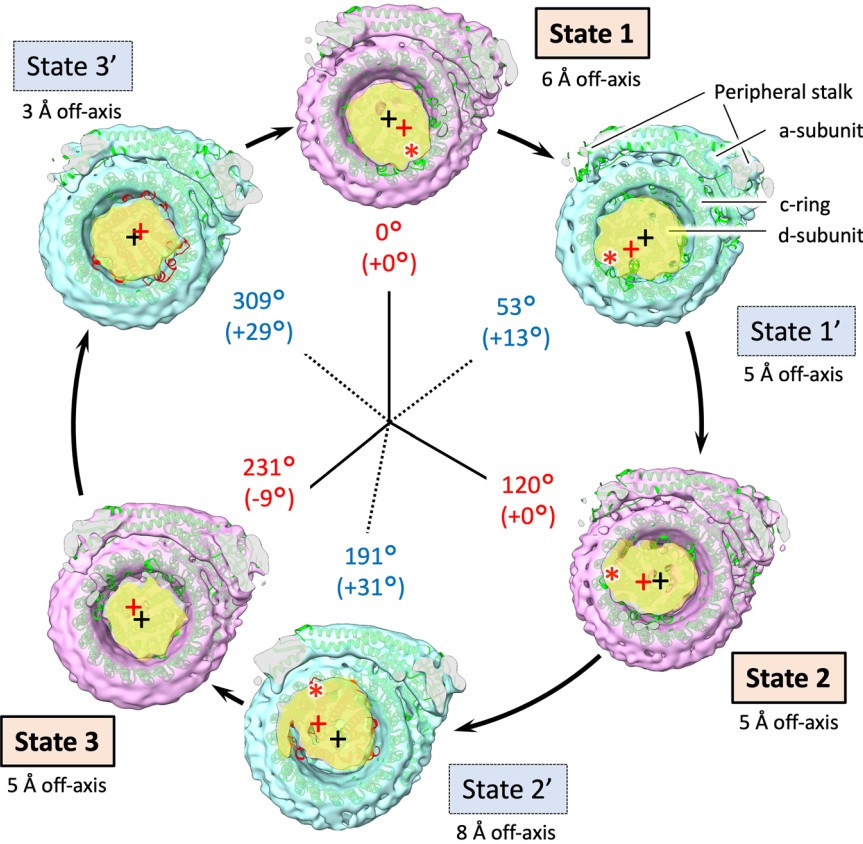

**Fig. 4 V$_o$ domain cross-section view of the six states.** The figure is laid out as in Fig. 2, viewed from V$_1$ to V$_o$ domains. The cross-sections of the d-subunit are displayed in yellow. The centre of the c-ring on each state is marked with a black cross, while the volume centre on the d-subunit is marked with a red cross. Deviation of the d-subunit from the centre of the c-ring (off-axis) is indicated by Å. Red asterisks show possible interaction points between the c-ring and the d-subunit. Cumulative rotation angles of the d subunit are labelled as the same as Fig. 3.

the EhV-ATPase. The rotor shaft terminus of the d-subunit and the c-ring in the V$_o$ domain showed relatively lower resolution, but when applying local-resolution filtering a density bridge was identified between the d-subunit and the extrinsic arm of the a-subunit (blue asterisk in State 3 in Fig. 5). The molecular interaction between these subunits may cause interference in the main pause of State 3 (Table 2). The subpause angle of the State 2' (+32° from the expected subpause angle) may also reflect the similar interference of the density bridge (blue asterisk in State 2' in Fig. 5). As previous work utilised only V$_1$ domain[15], steric hindrance from the membrane-extrinsic region of the a-subunit would not be observed. In this study, no significant distortion of the rotor was also observed during rotation (Table 2 and Supplementary Fig. 5) suggesting that the coupling between V$_o$ and V$_1$ is rigid, as reported by Otomo et al. using single-molecule imaging studies[45].

The off-axis rotor assembly was observed between the d-subunit and the c-ring (Fig. 4). We measured the distance between the centre of c-ring and the volume centre of the d-subunit (Fig. 4 and Table 2). The smallest gap distance of 3 Å observed in State 3' was gradually increased during turnover, and the largest gap distance of 8 Å appeared in State 2'. After that, the gap distance gradually decreased again towards the smallest 3 Å in State 3'. Off-axis motion may be related to the interference of the density bridge (Fig. 5). The off-axis distance from the c-ring centre with the off-axis motion increases when the interaction between the d-subunit and the c-ring moves to the opposite side of the a-subunit (Fig. 4, red asterisks in State 1 to State 2 in Fig. 5). However, the off-axis distance decreases after the interference between the d-subunit and the extrinsic arm of the

a-subunit (blue asterisks in Fig. 5 and Table 2). Finally, the off-axis motion was minimised at State 3' (Figs. 4 and 5). Perhaps the off-axis motion necessary for the interaction of the d-subunit and the c-ring is re-centred by the interference between the d-subunit and the extrinsic arm of the a-subunit. Supplementary Fig. 6 highlights the degree to which the rotor axis deviates from the vertical axis through the c-ring for each state, when viewed parallel to the lipid membrane from two perpendicular views.

State 2' is unique among the intermediate states and has only a small (global) resolution loss compared to the major states, while State 1' and State 3' are markedly lower resolution (>7 Å) (Supplementary Figs. 3 and 4 and Table 1). Despite containing a comparable number of particles to States 1' and State 3', State 2' is still <5 Å. In State 2', the d-subunit forming the rotor shaft terminus is approaching the extrinsic arm of the a-subunit (Fig. 5). This region is frustratingly the weakest point of many of the reconstructed maps, indicating that it is highly mobile. However, the State 2' reconstruction shows a potential interaction between the d-subunit and the a-subunit at this position when the map is filtered by local resolution (blue asterisk in State 2' in Fig. 5). This may contribute to the improved stability in addition to the potential re-centring of the rotor, and corresponding improvement in resolution of State 2' compared to State 1' or State 3' despite the relatively low particle count.

Based on these results and observations of the nucleotide binding pocket in V$_1$ domain (Fig. 6 and Supplementary Figs. 7–12) and those of the previous structural and single-molecule studies of isolated V$_1$ domain[15,34], we have built a catalytic model of EhV-ATPase through six states (Fig. 7). The process involved in a single rotation of the rotor within the V$_1$

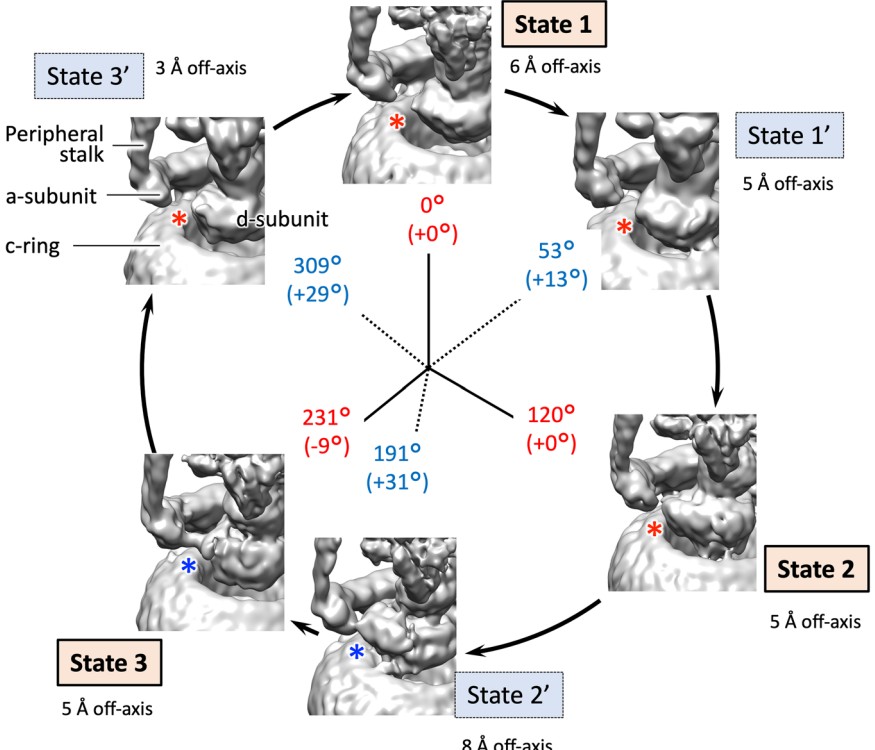

**Fig. 5 Isosurface view of the reconstructions for each state focussed on the rotor and terminus of the a-subunit arm.** The figure is laid out as in Fig. 2, viewed down from V$_1$ to V$_o$ domains. The maps were filtered by their local resolution. A red asterisk indicates no interaction (density bridge) between the rotor and the a-subunit arm, a blue asterisk indicates an interaction (density bridge). Rotational state of rotor indicated for each state. Total rotation of the rotor at F subunit, with the change from previous orientation in brackets, is labelled in red for the major states and blue for the intermediates. The off-axis distances shown in Fig. 4 are indicated externally of the circle.

domain consists of three repeats between three major states, each of which includes a phosphate release, an ATP binding, an ADP release, and an ATP hydrolysis event. These reactions would correspond to a conformational shift in the associated A/B dimer, which in turn would affect the rotation of the rotor. For example, looking at the event that occurred between State 1 and State 2, there is a clear distinction between the processes from State 1 to State 1' and State 1' to State 2. In the early process from State 1 to State 1', inorganic phosphate is released from the hydrolysed ATP (ADP + Pi) in the closed pocket, and another ATP binds to the open pocket. During this period, the structure of the pockets of "open", "closed", and "semi-closed" remains similar, and the rotor rotates 55°. Then, in the later process between State 1' and State 2, the ADP is released from the closed pocket and the structure of the pocket changes from "closed" to "open". The "semi-closed" pocket changes to "closed", and ATP is hydrolysed to ADP and Pi. The open pocket containing ATP becomes "semi-closed". During this period, the rotor further rotates 65°. A similar model can be applied to the process between State 3 and State 1, but the rotation angle of the rotor in V$_1$ domain is different: 75° (309° at State 3'–234° at State 3) between State 3 and State 3' and 51° (360° at State 1–309° at State 3') between State 3' and State 1 (Fig. 7). Differences in the rotor angle of the rotation can be caused by the rotor's off-axis assembly, which is clearly observed at the V$_o$ domain level (Fig. 4).

The special case of the model was observed between State 2 and State 3 (Fig. 7). The structure of the nucleotide binding pockets and the nucleotide densities of State 2' are like State 3 of the major state except for the rotor angle (Supplementary Figs. 10 and 11), which has not yet completed transition from State 2 to State 3. This structure indicates that the catalytic events for Pi release, ATP binding, ADP release, and ATP hydrolysis occur in the early

process between States 2 and State 2', and only rotor rotation occurs in the later process between State 2' and State 3. The abnormal behaviour can be caused by the interference between the d-subunit and the extrinsic arm of the a-subunit described above (Fig. 5). The interactions between these subunits that cause re-centring of the off-axis rotor between State 2' and State 3 may cause rotor rotation without chemical events. Unfortunately, while the presence or lack of nucleotide in a binding pocket is easy to determine, differentiating between ATP, ADP with inorganic phosphate and ADP without inorganic phosphate is considerably more difficult and will require extremely high resolutions in the V$_1$ domain. This mechanism is like that recently proposed by Shekhar et al.[53]. State 2' not only exhibits a state closer to that of the binding/catalytic dwell (major states in this study) than the ADP release dwell (intermediate states in this study), but the rotor has also advanced more than State 1' with a 74° (194° at State 2'–120° at State 2) rotation. State 3' shows a similarly advanced rotation of 75° (309° at State 3'–234° at State 3) when compared to State 1' but presents densities corresponding to three possible bound nucleotides. This indicates one of the densities is in the ADP release dwell and is thus consistent with the previously proposed model of turnover[15]. In Fig. 7, the hydrolysed ATP (ADP + Pi) in the closed states cannot be directly observed at current resolutions. In addition to our research group's single-molecule imaging results[34,45], MD simulations postulate an "ADP ·Pi-bound form (Pi-release dwell)" in this closed state[53]. In F$_1$-ATPase, this hydrolysed ATP in the closed states is defined as "half-open state"[40]. In contrast, the structures of the nucleotide binding pocket in States 1' and 3' are not clear at current resolution (>7 Å) (Supplementary Figs. 8 and 12). Nucleotides are thus temporarily assigned in the model (grey words in Fig. 7) according to the catalytic model proposed by the recent structural and single molecular analysis results[15,34]. Further

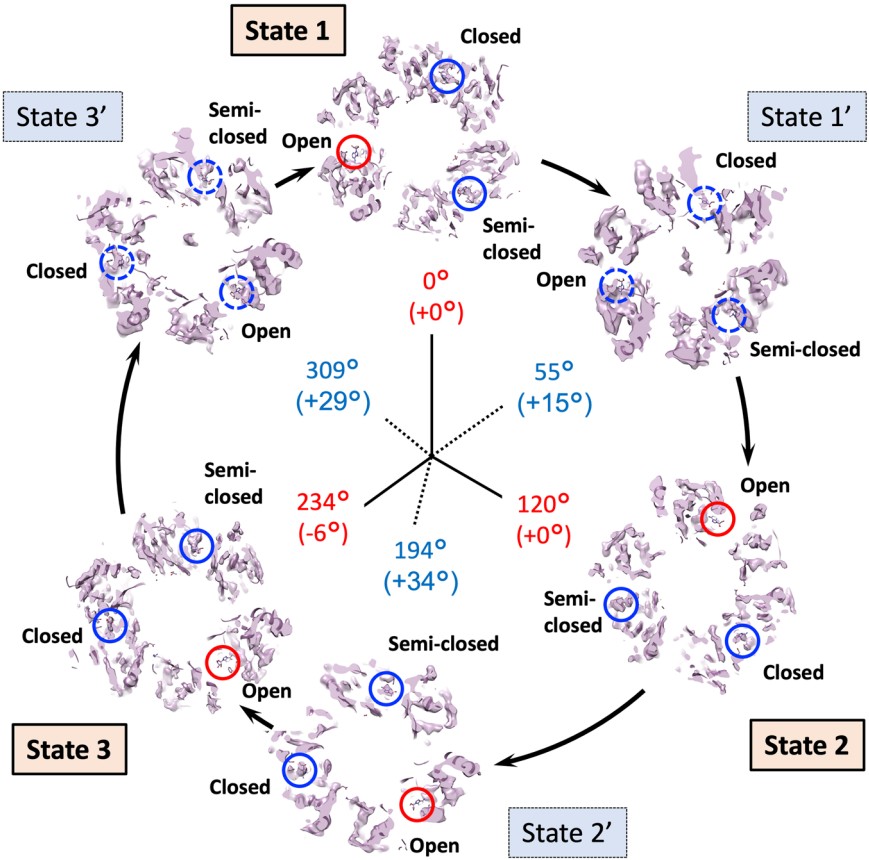

**Fig. 6 The nucleotide binding pocket densities of the different states in EhV-ATPase.** The figure is laid out as in Fig. 3, viewed down from V₁ to Vₒ domains. The maps are sliced through the density map at the level of the nucleotide binding pocket, and with PDBID:5KNC fitted to check whether density for a bound nucleotide is present or not. Blue circles indicate the presence of density corresponding to the bound nucleotide in the fitted PDB, red circles indicate missing density (only nucleotide from PDB is visible). Blue dotted circles in States 1' and 3' indicate the possible presence of nucleotide densities, as they are less clear at >7 Å resolution. Rotational state of rotor indicated for each state of six. Cumulative rotation angles of the rotor D subunit at the V₁ domain are labelled as the same as Fig. 3.

studies will elucidate the status of nucleotides in the pocket with higher resolution.

When the EhV-ATPase is activated, the relative position between the a-subunit and the c-ring changes continuously within the membrane (Fig. 8 and Table 2). Despite our best efforts, we have thus far been unable to improve the clarity of the Vₒ domain beyond that exhibited in the whole-complex reconstructions. We attribute this to the use of Na⁺, Mg²⁺ and ATP to generate turnover prior to vitrification, causing the c-ring to be present in multiple rotational states within the detergent, even within the distinct classes. It is possible to narrowly fit the α-helices in the density (which comprises most of the c-ring and a-subunit), but clarifying side-chain identity and orientation is essentially impossible (Fig. 8). Therefore, when examining the Vₒ domain, we simply measured the distance between the c-ring and the a-subunit from the Cα of the residues of interest: R573 and R629 for a-subunit, E139 for c-ring, where E139 is suggested to interact with R573 and R629 in the a-subunit (which are vertically aligned) to prevent ion transport without c-ring rotation[22,29,45]. However, because of the lower resolution in Vₒ domain, two possibilities for the E139 location must be considered. This is because E139 is positioned in one of two outward-facing transmembrane helices. Therefore, we measure both possible distances, labelled as E139a and E139b in Fig. 8. As a result, the major three states show the average distance of 9.7 ± 0.1 Å from the residue R573 of the a-subunit to the residue E139a, and the average distance of 12 ± 0.3 Å to the residue E139b (Fig. 8 and Table 2). These distances are approximately consistent

in all three major states, where one of the two helices locates closer to R573 of the a-subunit. The intermediate three states show that both helices are at more similar averaged distances: 11 ± 0.6 Å for the helix containing residue E139a and 11 ± 0.4 Å for the helix containing residue E139b. The result for R629 was similar. Previous reports showed that each monomeric c-ring member binds a single Na⁺ ion in a pocket consisting of the coordinating residues L61, T64, Q65, Y68, Q110 and E139[29], where Y68 coordinates with E139 rather than directly with bound Na⁺. Further, E139 plays a significant role in the binding of Na⁺ for translocation from one side to the other of the membrane[30] with mutagenic experiments to E139(D/Q) eliminating Na⁺ transport[27]. R573 of the a-subunit is proposed to act as a "latch" for c-ring rotation (to prevent c-ring rotation in the wrong direction, and thus a shortcut of the Na⁺ ion transport)[45,54]. The equivalent residue of the TtV/A-ATPase is R563 of the a-subunit and has been proposed to act to prevent this shortcut in *T. thermophilus*[22]. These observations suggest that in the major state both binding and release of transported Na⁺ positively occur between R573 of the a-subunit and E139 in the c-ring. By contrast, in the three intermediate states the c-ring would be in an intermediate stage of rotation, with no binding pocket directly accessible to either the ion ingress or egress pores.

In this study, we first report the structures of the six states of EhV-ATPase, which consisted of three major states and three intermediate states. EhV-ATPase showed a variety of rotor orientations in one major state and three intermediate states. These were suggested to be caused by the off-axis rotor

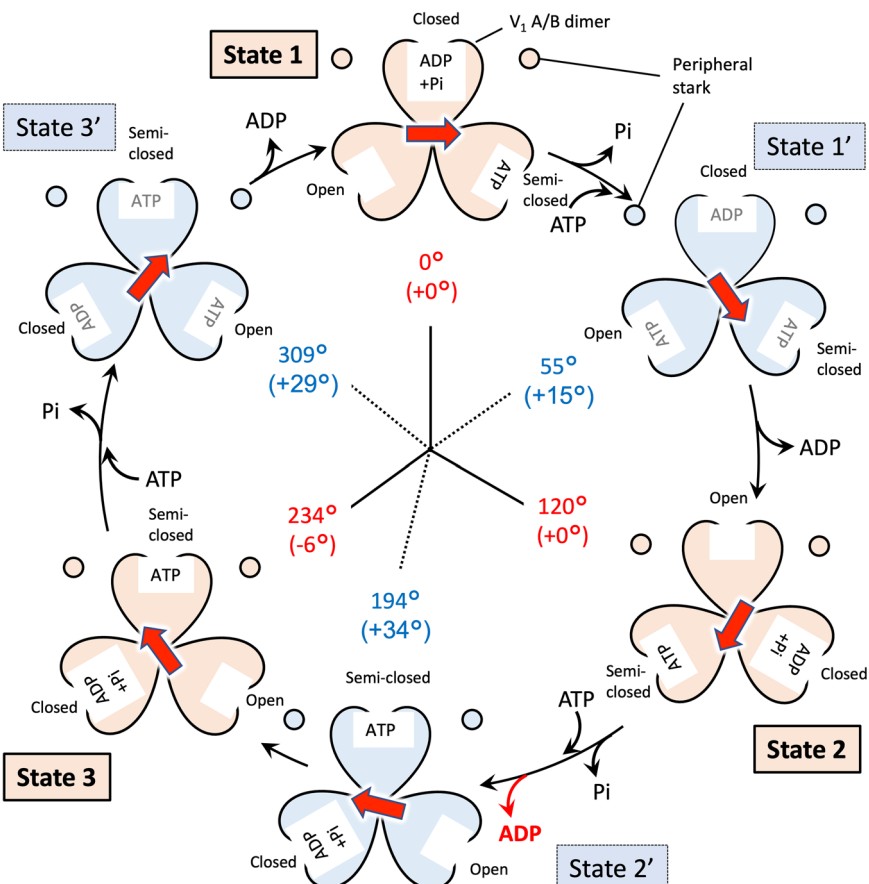

**Fig. 7 Schematic overview of the turnover of EhV-ATPase.** The $V_1$ domain is shown by the six states of the major and intermediate conditions, when viewed from $V_1$ to $V_o$ domains. Rotor D subunit orientation is indicated with a red arrow. In each lobe of the $V_1$ trimer is indicated whether the nucleotide binding pocket is empty, or containing ATP, ADP or ADP + Pi. Nucleotide densities in states 1′ and 3′ are not well defined at the current resolution and are therefore assigned based on previous work by Suzuki et al.[15] and labelled in grey. The figure is laid out as in Fig. 6.

assembly and its interreference with the extrinsic arm of the a-subunit. Two of the three intermediate state with few interactions between them showed relatively lower resolution, representing intrinsic structural instability. Densities in the nucleotide binding pockets in the motor $V_1$ domain showed the different chemical states between the major and intermediate states. Especially, State 2′ state show a unique state, in which the chemical state is like the upcoming State 3, but the rotor orientation shows that of a subpause. In addition to the asymmetric dual peripheral stalks, the mismatch between the relatively smaller d-subunit and the larger c-ring likely causes the unique off-axis assembly of the rotor complex in EhV-ATPase. The structures of the eukaryotic V-ATPases revealed that additional subunits in the c-ring and the symmetric triplet peripheral stalks fix the use of an on-axis rotor assembly even in the large c-ring[55]. By demonstrating features from both prokaryotic and eukaryotic V-ATPases, EhV-ATPase may show the evolutionary pathway of the ubiquitous membrane protein complex family of V-ATPases from prokaryotes to eukaryotes. In addition to the recent molecular simulations of the 4-step rotation mechanism based on the X-ray crystallographic models[53], further investigations are necessary to understand the complete molecular mechanism of the rotary ion pump and elucidate this relationship between V-ATPases.

## Methods

**Expression and purification of recombinant EhV-ATPase.** Expression and purification of recombinant EhV-ATPase were performed as previously described[41]. In brief, *E. hirae* V-ATPase was expressed recombinantly in

*Escherichia coli* and purified by affinity purification with a Ni+-NTA (nitrilotriacetic acid) column (Ni+-NTA Superflow; Qiagen, Hilden, Germany). The column was preconditioned with buffer consisting of 50 mM potassium phosphate, 100 mM KCl, 5 mM MgCl$_2$, 20 mM imidazole, 10% glycerol, 0.05% n-dodecyl-β-D-maltoside (βDDM) at pH 7.5. The column was washed, and protein was eluted with buffer containing 50 mM potassium phosphate, 100 mM KCl, 5 mM MgCl$_2$, 300 mM imidazole, 10% glycerol, 0.05% βDDM at pH 7.5. The purified protein was concentrated using an Amicon Ultra 100 K filter (Merck Millipore, Billerica, Massachusetts, USA) before further purification using Superdex 200 gel filtration (GE Healthcare, Little Chalfont, UK) preconditioned with 50 mM Tris-HCl, 5 mM MgCl$_2$, 10% glycerol, 0.05% βDDM at pH 7.5. The rotor-rotating activity of the EhV-ATPase in presence of ATP, Na$^+$, Mg$^{2+}$ was confirmed by inhibition of ATP hydrolysis of the $V_1$ domain by DCCD, an inhibitor of c-ring rotation in the $V_o$ domain[33].

**Cryo-EM grid preparation.** Purified EhV-ATPase was mixed with ATP and Na$^+$ to final concentrations of 0.13 mM (protein), 7 mM (ATP) and 100 mM (Na$^+$) before vitrification by plunge freezing using an Vitrobot Mark IV (Thermo Fisher Scientific, Hillsboro, Oregon, USA) onto Quantifoil R1.2/1.3 copper grids (Quantifoil Micro Tools GmbH, Großlöbichau, Germany) which had been glow discharged immediately beforehand.

**Cryo-EM data acquisition.** Micrograph movies were acquired on a JEOL CRYO ARM 300 microscope (JEOL Ltd., Tokyo, Japan) equipped with a Gatan K3 direct electron detector (Gatan Inc., USA) at a sampling scale of 1.01 Å/pixel using SerialEM (4.0) automation[46]. Fifty frames were recorded for each movie at low dose conditions for a total dose of 50 e−/Å$^2$. Three good grids were used for data acquisition. Micrographs were collected in three rounds; 19,250 micrographs in the first session and a further 15,978 micrographs collected in a second round. Due to strong orientation preference, a further round of data collection was carried out at 20° stage tilt and 30° stage tilt collecting a further 5032 and 3112 micrograph movies at those respective tilts. As the Gatan K3 suffered from significant gain drift during data acquisition, gain references were generated by summing raw movie frames using the cisTEM (1.0 beta)[56] programme

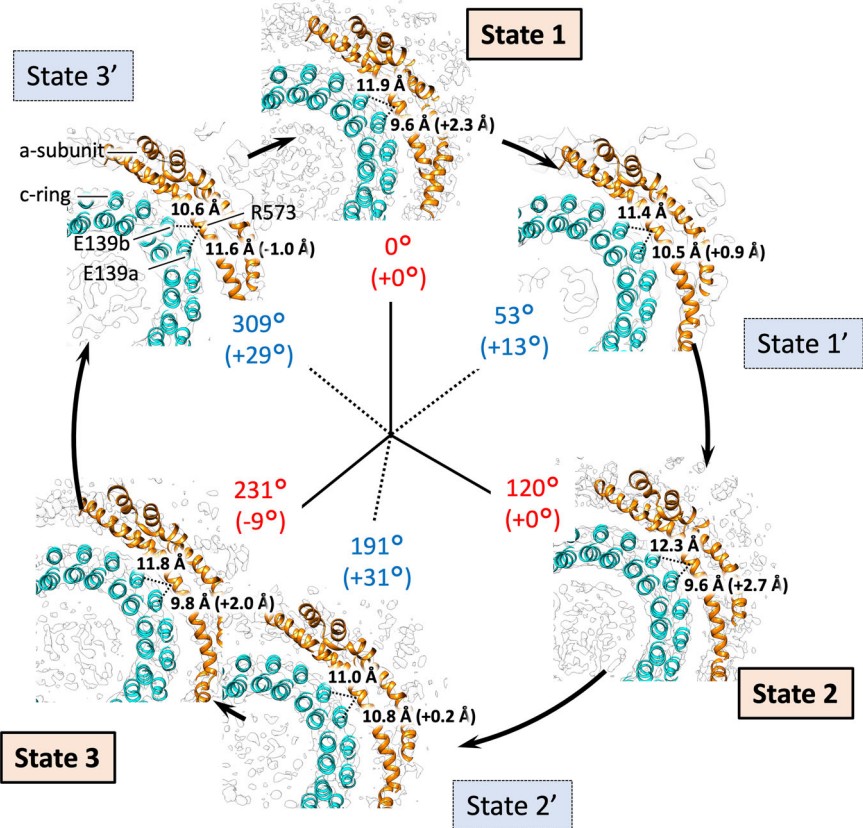

**Fig. 8 Focused view of the R573 (a-subunit)/E139a or E139b (c-ring) locale of the six states.** The figure is laid out as in Fig. 4, viewed down from $V_1$ to $V_o$ domains. Due to resolution constraints (Supplementary Figs. 3 and 4) it is not possible to reliably determine residue side-chain positions, thus distances are measured (shown with a dashed line) from Cα of E139a and E139b of the c-ring to Cα of R573 of the a-subunit. E139a and E139b are one of two possible E139 positions on the two outer transmembrane helices, respectively. Absolute distance for each state is indicated, with difference between the two in black brackets. Also see Table 2.

"sum_all_tif_files" in appropriate batches required to avoid poor gain correction. Movies were motion corrected using MotionCor2 (1.3.1)[57] and the contrast transfer function (CTF) estimated using CTFFIND (4.1.10)[58]. Data collection and image processing information are summarised in Table 1 and Supplementary Table 1.

**Cryo-EM data analysis.** A previously reported reconstruction[41] was used to generate projections for a first-round automated particle picking on a subset of micrographs with Gautomatch (0.53)[59]. These picked particles were extracted, 3× down-sampled and 2D classified. Clear classes were used for input into the RELION 3.1[47,49] autopicking routine. Due to difficulties visualising EhV-ATPase without phase plate[41], generous settings were used for autopicking resulting in a total of 4,383,273 particle candidates picked. Particles were picked from micrographs grouped into ~1000 randomised micrographs per group, particles extracted with 3× downsampling and 2D classified in batches. Each batch picked ~90–110,000 particles, and between 60–80,000 particles were carried forward per batch in "good" classes. The good classes from every five batches were grouped together and passed to 3D classification with an angular sampling of 7.5° into 15 classes, where classes which were evidently complete complexes were selected. Classes with a damaged $V_o$ (membrane) domain, or broken rotor shaft were discarded. Particles were combined and 3D refined with a soft mask to improve angular assignments, before further 3D classification with alignment disabled into 25 classes. These classes were selected and grouped based on the general position of the rotor before a further 3D refinement with a reference generated by "relion_reconstruct" using each particle selection and a mask generated from that reference, which further optimised the angular assignment. Further 3D classification was carried out on each set into 25 classes, and each class manually assigned to one of six states based on rotor position (State 1, 1', 2, 2', 3 or 3') (Supplementary Fig. 1). All 3D classification was carried out with the default $T$ value of 4.

A soft mask focussed on the $V_1$ domain and rotor shaft, excluding peripheral stalks and $V_o$ domain, was generated and a 3D refinement carried out once more. After this a 3D classification using the same mask was carried out into 25 classes with alignment off. This resulted in seven classes with estimated resolutions exceeding 8 Å, which were selected and classified once more into 25

classes. States 1 and 3, the two positions either side of State 3' were combined, and 3D classified once more into 15 classes with classes grouped based on rotor position. These class assignments were passed to individual 3D refinement. Particles were re-extracted and re-centred at full scale, with "relion_reconstruct" used to generate references of them with the maximum resolution set to 15 Å. Reconstructions of the complete complex in each state were carried out, with CTF refinement, resulting in final resolutions of shown in Table 1 and Supplementary Fig. 4. Localised masks were generated of the $V_o$ and $V_1$ domains, and further reconstructions carried out. This improved resolution for $V_1$ domain, but not for $V_o$ domain (Supplementary Figs. 3 and 4).

**Visualisation.** Micrographs, particles and 2D classes were visualised using the "relion_display" module of RELION 3.1[49]. 3D maps and PDB models were visualised using IMOD (4.9.12)[60], UCSF Chimera (1.16)[61] or UCSF ChimeraX (1.2)[62].

**Model building.** The $V_1$ domain crystal structures of *E. hirae* V-ATPase (PDBID: 3VR6)[14] and (PDBID: 5KNB, 5KNC)[15] were used for the $V_1$ domain, and the crystal structure of the c-ring (PDBID: 2BL2)[29] was used as a component of $V_o$. The a-, d-, E- and G-subunits were generated by homology modelling using the I-TASSER (5.1) suite[63], where the PDB models of 6C6L, 1R5Z, 4DT0, and 5GAS were selected by I-TASSER as templates, respectively. The homology models were fitted against independently generated cryo-EM maps (described above) of the varying states.

**Statistics and reproductivity.** The details about experimental design and statistics used in different data analyses performed in this study are given in the respective sections of "Results" and "Methods". For clearer comparisons between the states that describe how the subunits move relative to one another, we rigid body fit the molecular models of the c-ring and d-subunit to their corresponding densities, and calculate each centre of mass. After confirming a good rigid body fit of the d-subunit to the density, we used the UCSF Chimera "measure centre" function[61] for simulated maps of the c-ring and d-subunits on the same map grid. Using the centre of mass of the c-ring and d-subunit, we

calculated the X/Y offset relative to the central axis of the c-ring (Fig. 4 and Table 2).

**Reporting summary**. Further information on research design is available in the Nature Portfolio Reporting Summary linked to this article.

## Data availability

The cryo-EM maps of the entire EhV-ATPase complex have been deposited in the Electron Microscopy Data Bank under accession number EMD-34139 for State 1, EMD-34140 for State 1', EMD-34141 for State 2, EMD-34142 for State 2', EMD-34143 for State 3, and EMD-34144 for State 3'. The cryo-EM maps of the $V_1$ domain of the EhV-ATPase complex have been deposited in the Electron Microscopy Data Bank under accession number EMD-34145 for State 1, EMD-34146 for State 1', EMD-34147 for State 2, EMD-34148 for State 2', EMD-34149 for State 3, and EMD-34150 for State 3'. All other data are available from the corresponding author (or other sources, as applicable) on reasonable request.

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

## Acknowledgements

We thank Fumiaki Makino at JEOL, and Takayuki Kato at Osaka University, for helping data collection by CRYO ARM 300 (JEOL Inc.), and Akihiro Otomo at Institute for Molecular Science for helpful discussion. This study was supported by Interdisciplinary research project by young researchers in NINS (to C.S.), JSPS KAKENHI Grant Number JP16H06280 (to R.I.), JSPS KAKENHI Grant Number JP22H04926 (to K.M.), AMED BINDS under Grant Number JP17am0101001 and JP22ama121005j0001 (to K.M.), and the Grant-in-Aid for Scientific Research on Innovative Areas "Molecular Engine" (JP19H05380 to H.U., JP18H05425 to T.M., and JP18H05424 to R.I.).

## Author contributions

K.M. conducted and designed the experiments; H.U., T.M., and R.I. prepared the specimens. C.S. collected the cryo-EM data. R.N.B.S. process the cryo-EM images. R.N.B.S. and K.M. wrote the paper. All authors revised the paper.

## Competing interests

The authors declare no competing interests.
