## [Peer Review File · Communications Biology]

Reviewers' comments:

Reviewer #1 (Remarks to the Author):

The manuscript by Raymond et al described non-uniform rotor rotating by using the same grid preparations which identified six states of the EhV-ATPase complex with three major states at resolutions of 4.2-4.4 Å, and three intermediate states at resolutions of 4.8-7.7 Å. These intermediate states were shown to be subpause states, which is the first structural evidence for subpauses in V-ATPase. The authors concluded that during turnover the orientations of the rotor complex are not as perfectly uniform as expected, and the non-uniform rotor rotation is performed through the entire complex. Overall, the manuscript contains too much redundant information and should be reorganized before being accepted.

Major issue:

1. In figure 6 and S6, and considering the resolution, one can hardly confirm the presence of an ATP/ADP molecule in the state 1' open conformation. The density is unclear, and it could be argued that there is no substrate in there. Besides, the 1' "closed state" seems like semi-closed in figure 3.
2. F-subunit position, the rotor (D subunit) orientation, V1 domain and the ATP binding states all show that state 2' is similar to state 3 rather than state 2, so state 2' should be classified under state 3.
3. There is too much details in the introduction, especially from line 78-123. Most contents are not key to the story and should be deleted and simplified.

Minor issue:

Too many cosmetics and format problem there, such as in Figure. 5. Panels are overlapped with each other, table 1 defocus -1µM—2µm, etc.

Reviewer #2 (Remarks to the Author):

The researchers identified 6 intermediate states for the EhV-ATPase by cryo-EM and provide direct structural evidence for the existence of subpauses in V type ATPase. This is very interesting to bring us new views for the non-uniform rotor rotation of V-ATPase, and the only thing is two of these intermediate states show a relative lower resolution.

1. In line 62-64. For the classification of V1 and Vo, could you add some citations? Because for the V-ATPase in yeast and mammals, the EG subunits are always belong to V1 domain.
2. Due to the resolution limitation of V1 domain from the six states (resolution range from 3.8 Å - 7.7 Å), the small molecules couldn't be well confirmed. The researchers provided the density maps for the existence of these nucleotides, but some are not well fitted. How can you verify the ATP molecules? Like in Ref 19 in the paper, they found an unknown molecule in the ABsemi conformation. Can you provide more evidence for that? Furthermore information including the comparison of the binding sites of ATP/ADP with other EhV1 structures could be used to support it. Please add more explanation to address the verification of ATP.
3. Can you provide some evidence to show the activity of the Eh purified V-ATPase in presence of ATP, Na⁺, Mg²⁺? (For example, experiment or related paper)
4. For the Figure 8, the Vo resolution is low as mentioned, so the measurement may not be accurate, how can you assign the corresponded helices? Could you see the relative position of the helix bundles of these states, especially for state 1' and state 3'?

Reviewer #3 (Remarks to the Author):

Burton-Smith et al. propose a rotary catalytic scheme for the intact *Enterococcus hirae* V-type ATPase, describing off-axis rotation and an interaction between the rotary and stationary parts. These findings are novel, though related to recently published work <https://doi.org/10.1021/acscentsci.1c01599>. The findings would be of interest to the rotary ATPase community.

Improvements required before publication:

(i) Overall, I found the manuscript confusing with a drawn-out discussion. I suggest the authors try to simplify the text and present clear concise conclusions.

(ii) An opening supplementary figure describing what the authors mean by an "off-axis rotor" would greatly increase the clarity of the manuscript.

(iii) The red boxes in Figure 3 alone do not show that the D-subunit is rigid between the V1 domain and F-subunit. Could the authors provide images showing superposition of the maps for the rotor in various rotary positions to show that there's no movement in this region? Is the off-axis movement being mediated by the interface between the d-subunit and c-ring?

(iv) Further to point (iii), it was unclear to me how the d-subunit able to move relative to the c-ring. What interactions allow such a mobile interaction? It doesn't look like the c-ring deforms. Is this just an intrinsic hinge between the d-subunit and c-ring? If so, this fact was lost on me due to the explanation of the change in centre of subunits. A diagram showing such a movement would aid the manuscript substantially.

(v) How was the centre of the d-subunit calculated?

(vi) Could the authors expand on the evidence for the nucleotide/Pi occupancy suggested in Figure 7. These occupancies appear different to other published structures of F- and V-ATPases in similar rotatory states (e.g. ADP+Pi bound to a closed state). One assumes the occupancy is based on data within reference 40, as they do state it would be challenging to identify nucleotides at this resolution.

(vii) PDB validation reports weren't provided for any of the maps. These are required to review this manuscript.

(viii) Why was a concentration of 7 mM ATP and 100 mM Na⁺ chosen for this study? Is this physiologically meaningful? Are the authors confident that the enzyme is rotating at the point of freezing? Other groups have attempted similar studies on related intact enzymes (DOIs: 10.7554/eLife.43864.002 and 10.1038/s41467-022-29893-2) so it would be important to compare the methods used to here.

(ix) Methods: it would be useful to include the t value used during the 3D classification.

(x) Citation in line 494 isn't a number.

(xi) Line 499: is it possible to include which structures were used by I-TASSER to make the homology models.

Responses are labeled in blue. Page and line numbers are based on the **marked** file.

Reviewers' comments:

Reviewer #1 (Remarks to the Author):

The manuscript by Raymond et al described non-uniform rotor rotating by using the same grid preparations which identified six states of the EhV-ATPase complex with three major states at resolutions of 4.2-4.4 Å, and three intermediate states at resolutions of 4.8-7.7 Å. These intermediate states were shown to be subpause states, which is the first structural evidence for subpauses in V-ATPase. The authors concluded that during turnover the orientations of the rotor complex are not as perfectly uniform as expected, and the non-uniform rotor rotation is performed through the entire complex. Overall, the manuscript contains too much redundant information and should be reorganized before being accepted.

Thank you for your encouraging comment. We have reorganized the manuscript to remove the redundant information. For example, we have merged the 2nd and 3rd paragraphs in the discussion section, because the rotation gaps in main and sub pauses are both suggested to be caused by the interference between the off-axis rotor and the stator subunits. The modifications are left in the marked file.

Major issue:

1. In figure 6 and S6, and considering the resolution, one can hardly confirm the presence of an ATP/ADP molecule in the state 1' open conformation. The density is unclear, and it could be argued that there is no substrate in there. Besides, the 1' "closed state" seems like semi-closed in figure 3.

As you suggest, it is difficult to directly identify whether a nucleotide is present in each ATP binding pocket of the state 1', because it showed the worst resolution of 7.7 Å. Notably, the nucleotide density in the open conformational is slightly shifted from the position expected from the X-ray crystal model (Fig. S6). Also, it is difficult to determine whether it is closed or semi-closed from its structure in the state 1' (Fig. 3). Therefore, we finally and globally concluded the conformations of three A/B dimers and the nucleotide states of State 1' in the six states. Toned down the discussion a bit by including an explanation in the main text as follows.

"In the intermediate states, despite the relatively low resolutions of the V1 domain (4.2 Å to 7.7 Å), we sought to find the nucleotide density in the ATP binding pocket. States 1' and 3' appear to contain densities in all three nucleotide binding pockets (blue circles in State 1' and 3' in Fig. 6, Table 3), while State 2' shows the similar binding coordination as State 3 (State 2' in Fig. 6, Table 3). These images are magnified in Figs, S7, S9, and S11, respectively. If these observations are true, the result is consistent with the notion that the major three states correspond to the main pauses waiting for ATP binding, observed in the previous single molecule study of isolated V1 domain." P. 11, Line 287 (Line 240 in clean file).

2. F-subunit position, the rotor (D subunit) orientation, V1 domain and the ATP binding states all show that state 2' is similar to state 3 rather than state 2, so state 2' should be classified under state 3.

Thank you for your suggestion. This is a point we discussed before submission. We followed the definition of states from other reports (e.g.: Nakanishi *et al.* 2019). As the rotor will only rotate in one direction, we denoted states which are intermediate as a subset of the previous conformation. Thus, until the rotor fully reaches the state 3 position, any orientation is denoted as an intermediate of states 2 and 3, thus, [2']; if there had been others, [2''], or [2'''] would have been used, also. We could, potentially, use "pre"-state to define it, but the question is at what point would you differentiate from "post"-last-state and "pre"-next-state? Here, we used the previous definition and discussed why the state 2' is similar to the state 3 rather than the state 2.

3. There is too much details in the introduction, especially from line 78-123. Most contents are not key to the story and should be deleted and simplified.

We have attempted to simplify the introduction. The modifications are left in the marked file.

Minor issue:

Too many cosmetics and format problem there, such as in Figure. 5. Panels are overlapped with each other, table 1 defocus -1 μ M—2 μ m, etc.

We have fixed all formatting errors we have found. The modifications are left in the marked file. Thank you for pointing two out.

Reviewer #2 (Remarks to the Author):

The researchers identified 6 intermediate states for the EhV-ATPase by cryo-EM and provide direct structural evidence for the existence of subpauses in V type ATPase. This is very interesting to bring us new views for the non-uniform rotor rotation of V-ATPase, and the only thing is two of these intermediate states show a relative lower resolution.

Thank you. Yes, we were likewise frustrated with the lower resolution of two of the intermediates and spent significant time attempting to improve them further. Unfortunately, this was unsuccessful. The fact suggests the inherent structural instability. We include this in main text as follows.

“Two of the three intermediate state with few interactions between them showed relatively lower resolution, representing intrinsic structural instability.” P. 18, Line 559 (Line 390 in clean).

1. In line 62-64. For the classification of V1 and Vo, could you add some citations? Because for the V-ATPase in yeast and mammals, the EG subunits are always belong to V1 domain.

Thank you for pointing this out. It was our misunderstanding. We have fixed the sentence as follows.

“V-ATPases are formed from two domains: the Vo and V1 domains. In the V-ATPase from *Enterococcus hirae* (EhV-ATPase), the V1 domain in cytoplasm consists of A-, B-, D-, E-, F-, G-, and d-subunits, and the Vo domain bound membrane consists of a- and c-subunits (Fig. 1).” P. 3, Line 62 (Line 60 in clean).

We have also newly included the reference 13.

2. Due to the resolution limitation of V1 domain from the six states (resolution range from 3.8 Å - 7.7 Å), the small molecules couldn't be well confirmed. The researchers provided the density maps for the existence of these nucleotides, but some are not well fitted. How can you verify the ATP molecules? Like in Ref 19 in the paper, they found an unknown molecule in the ABsemi conformation. Can you provide more evidence for that? Furthermore, information including the comparison of the binding sites of ATP/ADP with other EhV1 structures could be used to support it. Please add more explanation to address the verification of ATP.

Currently, the cryo-EM map resolutions are relatively low to directly identify the nucleotide densities. Therefore, the X-ray crystal model of 3 ADP-bound V1 complex was fitted into the cryo-EM maps to verify the presence of nucleotide molecules in the ATP binding pocket. The magnified figures are attached as Fig. S6 to S10. Added the following sentence to the main text to clarify this.

“In Fig. 6, the crystallographic model of the three ADP-bound V1 complex (PDBID: 5KNC) 15 was fitted into the cryo-EM maps to see whether density for a bound nucleotide is present or not” P. 11, Line 279 (Line 232 in clean).

3. Can you provide some evidence to show the activity of the Eh purified V-ATPase in presence of ATP, Na⁺, Mg²⁺? (For example, experiment or related paper)

In single molecule studies, EhV-ATPase shows a rotor rotation in the presence of ATP, Na⁺ and Mg²⁺ (Otomo *et al.* 2022, Iida *et al.* 2019). Our previous work (Tsunoda *et al.* 2018) showed that the expressed and purified recombinant EhV-ATPase complex was inhibited ATP hydrolysis of the V1 domain by DCCD, an inhibitor of c-ring rotation in the Vo domain. Furthermore, the EhV-ATPase complexes inserted a peptide tag into the rotor D-subunit showed different states with and without Fab binding upon the complex activation in the presence of ATP, Na⁺, Mg²⁺. We have added these explanations in the main text as follows.

“7 mM ATP and 100 mM Na⁺ were used to detect as many possible EhV-ATPase states during turnover. In single molecule studies, EhV-ATPase exhibits rotor rotation in the presence of these concentrations of ATP and Na⁺ 41,50. Our previous work showed that the EhV-ATPase complexes with peptide tag inserted into the rotor D-subunit exhibited different states with and without Fab binding upon activation of the complex with these concentrations of ATP and Na⁺.” P. 6, Line 165 (Line 131 in clean).

“The rotor-rotating activity of the EhV-ATPase in presence of ATP, Na⁺, Mg²⁺ was confirmed by inhibition of ATP hydrolysis of the V1 domain by DCCD, an inhibitor of c-ring rotation in the Vo domain.” P. 19, Line 599 (Line 421 in clean).

4. For the Figure 8, the Vo resolution is low as mentioned, so the measurement may not be accurate, how can you assign the corresponded helices? Could you see the relative position of the helix bundles of these states, especially for state 1' and state 3'?

We discussed this before submission and felt that not overlaying the density maps was clearer. However, we have now overlaid the density maps which show weak signals for the helices. Using UCSF Chimera, if we roughly place the c-ring crystal structure PDB into the Vo density and apply the “fit to map” function, helices are fitted to the same positions regardless of minor variations in orientation. Major variations will align incorrectly, as the “fit in map” function can struggle if the map and model are too greatly mis-aligned. We have modified the figure 8 to make the location of the helices clearer on the cryo-EM maps. However, as you suggested, the assignments of the helices are not exact, especially for the state 1' and state 3'. Thus, the statement has been slightly toned down as follows.

“It is possible to narrowly fit the α -helices in the density (which comprises most of the c-ring and a-subunit), but clarifying side-chain identity and orientation is essentially impossible (Fig. 8).” P. 16, Line 497 (Line 359 in clean).

Reviewer #3 (Remarks to the Author):

Burton-Smith *et al.* propose a rotary catalytic scheme for the intact *Enterococcus hirae* V-type ATPase, describing off-axis rotation and an interaction between the rotary and stationary parts. These findings are novel, though related to recently published work <https://doi.org/10.1021/acscentsci.1c01599>. <<https://doi.org/10.1021/acscentsci.1c01599>> The findings would be of interest to the rotary ATPase community.

Thank you for your valuable information. We have cited the paper in the main text as follows.

“In addition to the recent molecular simulations of the 4-step rotation mechanism based on the X-ray crystallographic models 61, further investigations are necessary to understand the molecular mechanism of the rotaru ion pump and elucidate this relationship between V-ATPases.” P. 18, 571 (Line 402 in clean).

Improvements required before publication:

(i) Overall, I found the manuscript confusing with a drawn-out discussion. I suggest the authors try to simplify the text and present clear concise conclusions.

We have tried to clean up the discussion. For example, we merged the discussion of rotation angle variations of the off-axis rotor, which are previously placed in the 2nd and 3rd paragraphs, into the result section. The modifications are left in the marked file.

(ii) An opening supplementary figure describing what the authors mean by an “off-axis rotor” would greatly increase the clarity of the manuscript.

We cited Tsunoda *et al.* 2018 (ref. 41) in P. 5 Line 103 (Line 106 in clean), and added an inset figure in Figure 1 for increasing the clarity.

(iii) The red boxes in Figure 3 alone do not show that the D-subunit is rigid between the V1 domain and F-subunit. Could the authors provide images showing superposition of the maps for the rotor in various rotary positions to show that there’s no movement in this region? Is the off-axis movement being mediated by the interface between the d-subunit and c-ring?

We believe this to be the case, yes. If we overlay the different rotations of the DF complex (using Matchmaker in UCSF Chimera) there is very little exhibited flexibility. Since we can’t see any deformation of the DF complex or the c-ring, we conclude that the d-subunit must be acting as a more flexible link. We have added a new supplementary figure 5.

(iv) Further to point (iii), it was unclear to me how the d-subunit able to move relative to the c-ring. What interactions allow such a mobile interaction? It doesn’t look like the c-ring deforms. Is this just an intrinsic hinge between the d-subunit and c-ring? If so, this fact was lost on me due to the explanation of the change in centre of subunits. A diagram showing such a movement would aid the manuscript substantially.

This information is included Figure 4. In Figure 4, the center of c-ring and the center of the d-subunit were indicated and the gap distances were labeled for each state. The interaction residues between these subunits were not clear at the limited resolution of the Vo domain. We have added the following sentence to make it clearer in the main text.

“However, the d-subunit was able to move relative to the c-ring (Fig. 4), causing off-axis rotation of the rotor.” P. 10, Line 248 (Line 209 in clean).

(v) How was the centre of the d-subunit calculated?

The centres of the d-subunit were determined from the centers of the concentric circles fitted to the map slices in Figure 4. We added the explanation in the main text

“Slicing these maps horizontally at a Vo domain position and measuring the distance from the centre of the c-ring to the concentric centre of the d-subunit cross-section, the off-axis centres of d-subunit were detected as 7.1 and 9.4 Å in State 2 and State 2’, while the smaller off-axis centres of the d-subunit were detected as 3.1 Å in State 3, and 2.0 Å in State 3’.” in P. 10, Line 253 (Line 214 in clean).

(vi) Could the authors expand on the evidence for the nucleotide/Pi occupancy suggested in Figure 7. These occupancies appear different to other published structures of F- and V-ATPases in similar rotatory states (e.g. ADP+Pi bound to a closed state). One assumes the occupancy is based on data within reference 40, as they do state it would be challenging to identify nucleotides at this resolution.

Currently we believe that there is no direct structural evidence for ADP and Pi, even other F- and V-ATPases. Our catalytic model in Figure 7 is based on our cryo-EM map and single-molecule imaging results by Iida et al. 2019 (former Ref. 40). The similar ATP hydrolyzed condition was also suggested by MD simulation in Shekhar et al. 2021. In F1-ATPase, it was defined as half-open state by Sobti et al. 2021. We have added sentences to explain these facts as follow.

“In Fig. 7, the hydrolyzed ATP (ADP + Pi) in the closed states cannot be directly observed at current resolutions. In addition to our research group’s single-molecule imaging results, MD simulations postulate an “ADP•Pi-bound form (Pi-release dwell)” in this closed state. In F1-ATPase, this hydrolyzed ATP in the closed states is defined as “half-open state.” in P. 16 Line 486 (Line 348 in clean).

(vii) PDB validation reports weren’t provided for any of the maps. These are required to review this manuscript.

In this study, we didn’t upload any PDB models, only cryo-EM maps. Previous reported PDB models (2BL2 for c-ring, 3VR6 for the AMP-PNP bound V1 complex, 5KNB for the 2 ADP-bound V1 complex, and 5KNC for the 3 ADP-bound V1 complex, 1R5Z for the d-subunit, and 6C6L for the a-subunit, 4DT0, and 5GAS for the EG-subunit) were used for the map annotations. Other than rotor orientation, no significant variation was shown from previous PDB models of different states, and the density for stalks was too weak to fit reliably.

(viii) Why was a concentration of 7 mM ATP and 100 mM Na⁺ chosen for this study? Is this physiologically meaningful? Are the authors confident that the enzyme is rotating at the point of freezing? Other groups have attempted similar studies on related intact enzymes (DOIs: 10.7554/eLife.43864.002 and 10.1038/s41467-022-29893-2) so it would be important to compare the methods used to here.

Thank you for this interesting point. The breadth of biological variation and adaptability is astonishing. However, we have found precise concentrations to be physiologically irrelevant in our groups paper below.

Iida, T. et al. Single-molecule analysis reveals rotational substeps and chemo-mechanical coupling scheme of *Enterococcus hirae* V1-ATPase. *J Biol Chem* 294, 17017–17030 (2019).

Otomo, A. et al. Direct observation of stepping rotation of V-ATPase reveals rigid component in coupling between Vo and V1 motors. *Proc Natl Acad Sci U S A* 119, e2210204119 (2022).

However, under these conditions we know the enzyme is active and rotating. For single molecule studies very high Na⁺ conditions are required to prevent ion binding from becoming rate limiting. Since these concentrations are used to show the six states in single molecule work, we decided introducing further variables was unwise.

If we use rate-limiting concentrations, we may isolate fewer states - or possibly more states! Although we are trying this right now with some biochemical difficulties. In ATPase assay, 25mM Na⁺ seems to get maximum rate, and V_{max} is the same at 100mM, so excess is not causing a problem. V_{max} for ATP is ~1mM, but 7mM does not cause an issue. Basically, excess isn’t a problem but guarantees that ion and substrate concentrations are not possibly rate limiting.

We have added explanations as follows in results section.

“7 mM ATP and 100 mM Na⁺ were used to detect as many possible EhV-ATPase states during turnover. In single molecule studies, EhV-ATPase exhibits rotor rotation in the presence of these concentrations of ATP and Na⁺. Our previous work showed that the EhV-ATPase complexes with peptide tag inserted into the rotor D-subunit exhibited different states with and without Fab binding upon activation of the complex with these concentrations of ATP and Na⁺.”, P. 6 Line 165 (Line 131 in clean).

(ix) Methods: it would be useful to include the t value used during the 3D classification.

P. 21, Line 645 (Line 467 in clean): We have included the T value, which was 4 (the default). We used this strategy for two reasons: first, given the range of resolutions, we were concerned about potential overfitting, which can be an issue with higher T values. Second, with some early testing using a higher T value did not separate rotor orientations well.

(x) Citation in line 494 isn't a number.

P. 22 Line 664 (Line 486 in clean): It has been fixed to ref. 60.

(xi) Line 499: is it possible to include which structures were used by I-TASSER to make the homology models.

We have modified the sentence as follows.

“The a-, d-, E- and G-subunits were generated by homology modelling using the I-TASSER suite 61 , where the PDB models of 6C6L, 1R5Z, 4DT0, and 5GAS were selected by I-TASSER as templates, respectively.”, P.22 Line 669 (Line 491 in clean).

Reviewers' comments:

Reviewer #1 (Remarks to the Author):

This study have provided new insights into the mechanisms of this important enzyme and have effectively addressed all of my concerns. I recommend that this study be published.

Reviewer #2 (Remarks to the Author):

The authors addressed most of my concerns, but for question 2, my point is these supplementary figures perhaps only suggest the presence or absence of these nucleotides. For S1, S2, S3, the assignment of nucleotides is well known as lots of paper stated, but for all the subpause states, the assignment for ATP or ADP should be careful. I want to know whether you can see the clear secondary structure near these nucleotide binding sites? Whether you can see the clear binding motif, like walker A motif? Whether these motifs can form the ATP/ADP binding circumstance as reported papers? The detailed information is lacked. And I just want some comparisons and citations between these states with the classic AB dimers. Even though some local resolutions are low, the binding sites may be stable and provide some information. The authors said something about the similar binding coordination and assignment, but their supplementary figures didn't clearly show this. This information can provide indirect evidence to support the correct assignment for the nucleotides.

Reviewer #3 (Remarks to the Author):

(i) The manuscript is much clearer now that the authors have included the inset to describe the "off-axis rotor". However, I suggest lines 111-112 in the introduction are edited to include other systems. Currently these lines read: "The off-axis nature of the rotor has thus far only been identified in EhV-ATPase". It is this reviewer's understanding that off-axis rotors have been identified in other organisms (I have seen others in the literature, but I could easily find these figures for yeast [Fig. 14b in DOI: 10.1098/rstb.2000.0589] and bacterial [Fig. 2 in DOI: 10.1038/ncomms1693]).

(ii) Supplementary Figure 5 shows superimposed models rather than maps for the D-subunit. Figure 4 attempts to show the difference between the centre of the d-subunit and centre of the c-ring. However, it is still unclear to this reviewer what is changing between the states and what is mediating/allowing the subunits to move relative to one another. The authors need to show clear comparisons between the states that describe how the subunits move relative to one another. For example, can the authors align all states (preferably maps) to the c-ring and show a side view of the d/D/F subunits flexing back and forth? One would expect to observe the subunits rocking back and forth.

(iii) The explanation of how the centres of the subunits were measured reads as a poor inaccurate method. How could the authors ensure that the c-ring axis was perpendicular to the horizontal slice? Even if the authors could do this, if the d-subunit rotates perpendicular to the c-ring, then using a horizontal slice perpendicular to that used to slice the c-ring will induce an artificial translation (as the d-subunit is not a sphere). This gives the impression of a translation when it should be identified as a rotation. At any rate, using an accuracy of 0.1 Å to describe relative changes in the centre of these subunits seems far too high. A potentially better method would be to rigid body fit the subunits and calculate the centre of mass of the c-ring and d-subunit. However, to make the comparison Figure (as in Figure 4), the states will need to be aligned so that all the c-ring axes are perpendicular to the plane of the figure. Otherwise the differences between the centre of masses will change due to the viewing angle, rather than changes in the subunit positions.

(iv) I requested the PDB validation reports for the maps, not for models. If the authors prefer, they could upload the raw maps with the submission?

First, I would like to apologise to the reviewers and editor for the delay in resubmission. We have carefully read and responded to the reviewers' comments and revised the manuscript reflecting their all suggestions. Author responses are displayed in blue.

Reviewers' comments:

Reviewer #1 (Remarks to the Author):

This study have provided new insights into the mechanisms of this important enzyme and have effectively addressed all of my concerns. I recommend that this study be published.

Thank you for your positive feedback!

Reviewer #2 (Remarks to the Author):

The authors addressed most of my concerns, but for question 2, my point is these supplementary figures perhaps only suggest the presence or absence of these nucleotides. For S1, S2, S3, the assignment of nucleotides is well known as lots of paper stated, but for all the subpause states, the assignment for ATP or ADP should be careful.

Thank you for your critical examination. We absolutely agree. Of the intermediate states, only state 2' is clear, as the resolution in the binding pockets is sufficient to identify a small density which cannot be attributed to the protein (new Fig S10). For the states of 1' and 3', focussed views of the distinguishable binding sites are shown in new Figs. S8 and S12. The inset in new Fig. S8 is an example of the difficulty for states 1' and 3': even at 9 σ , density for the nucleotide is a large blob. Therefore, these nucleotide statuses of their binding pockets have been labelled with blue dotted circles in the new Figs. 6, S8, and S12.

I want to know whether you can see the clear secondary structure near these nucleotide binding sites? Whether you can see the clear binding motif, like walker A motif? Whether these motifs can form the ATP/ADP binding circumstance as reported papers? The detailed information is lacked. And I just want some comparisons and citations between these states with the classic AB dimers.

Even though some local resolutions are low, the binding sites may be stable and provide some information. The authors said something about the similar binding coordination and assignment, but their supplementary figures didn't clearly show this. This information can provide indirect evidence to support the correct assignment for the nucleotides.

Thank you for your suggestion. In states 1, 2, 3 and 2', we can clearly see density in the binding pocket. Therefore, we have included the enlarged views of the nucleotide binding pockets in each supplementary figures S7, S9, S10, and S11. When we rigid body fit PDBs 5KNB/5KNC into density (depending on hypothetical binding state) we can observe density which the lysine and threonine residues of the Walker A motif (G-x-x-x-G-K-T) occupy. As

glycine is the smallest amino acid side chain, observing density for it is effectively impossible at these resolutions. We have added a sentence in the main text (P. 11, Line 14).

For states 1' and 3', the binding sites are not appreciably higher resolution than the rest of the reconstruction (Figs. S8, S12) at around 7 Å. While there is density which can be assigned to something which is not occupied by protein from previously reported PDB models, precise identification is impossible (see inset in Fig. S8). For this reason, we assign ADP/ATP based on previous, higher resolution crystallographic reports of just the V1 domain (Suzuki et al., 2016). The model is also based on the recent single molecule work initially proposing the intermediate states (Iida et al., 2019). We have added the explanation in the main text (P. 16, Line 20). In the catalytic model of Figs 7, the possible nucleotides in states 1' and 3' have been labelled in grey.

Reviewer #3 (Remarks to the Author):

(i) The manuscript is much clearer now that the authors have included the inset to describe the “off-axis rotor”. However, I suggest lines 111-112 in the introduction are edited to include other systems. Currently these lines read: “The off-axis nature of the rotor has thus far only been identified in EhV-ATPase”. It is this reviewer’s understanding that off-axis rotors have been identified in other organisms (I have seen others in the literature, but I could easily find these figures for yeast [Fig. 14b in DOI: 10.1098/rstb.2000.0589] and bacterial [Fig. 2 in DOI: 10.1038/ncomms1693]).

Thank you for your kind suggestion. When describing the *E. hirae* V-ATPase, the term “off axis rotor” is used to describe the fact that the rotor is not centred on the centre of the C-ring. This is caused by the mismatch of the prokaryotic-style stalks but large, eukaryote-style c-ring, with 40 transmembrane helices. Other (V)-ATPases may have the whole rotor/c-ring slightly offset relative to the axis of the A3B3 pseudo-hexamer, but none we have found in the literature except for *E. hirae* V-ATPase have this mismatch in stalk/c-ring arrangement. For *E. hirae*, the c-ring is “on axis” relative to the V1 domain, while in [DOI: 10.1098/rstb.2000.0589] the rotor/c-ring as a complete unit is “off axis” and in [10.1038/ncomms1693] the D-subunit is “off axis”, but the d-subunit is still centred in the smaller, 24 transmembrane helix c-ring. Therefore, we have added a sentence to make this clear in P. 5 Line 22 with these references.

(ii) Supplementary Figure 5 shows superimposed models rather than maps for the D-subunit. Figure 4 attempts to show the difference between the centre of the d-subunit and centre of the c-ring. However, it is still unclear to this reviewer what is changing between the states and what is mediating/allowing the subunits to move relative to one another. The authors need to show clear comparisons between the states that describe how the subunits move relative to one another. For example, can the authors align all states (preferably maps) to the c-ring and show a side view of the d/D/F subunits flexing back and forth? One would expect to observe the subunits rocking back and forth.

Thanks for the detailed instructions. We have updated this appropriately and have included diagrams using the maps as new Figure S6. In these maps, the position of the c-ring was

fixed. As a result, the v1 domain was observed swayed back and forth (Fig. S6).

(iii) The explanation of how the centres of the subunits were measured reads as a poor inaccurate method. How could the authors ensure that the c-ring axis was perpendicular to the horizontal slice?

Thank you for your suggestion. We have tried again, using the PDB model of 2BL2 to align the maps, then using the “molmap” function in UCSF Chimera to create a volume of just the d-subunit. Without the D/F complex, the d-subunit has a cup (or socket) within which the D/F complex interacts. Measuring from the deepest recess of this socket, the offset from the centre of the c-ring to the d-subunit does not deviate appreciably from our original estimates (although it does a little). We have added explanations in P. 10, Line 13 and in the top and bottom rows of new Figure S6.

Even if the authors could do this, if the d-subunit rotates perpendicular to the c-ring, then using a horizontal slice perpendicular to that used to slice the c-ring will induce an artificial translation (as the d-subunit is not a sphere). This gives the impression of a translation when it should be identified as a rotation. At any rate, using an accuracy of 0.1 Å to describe relative changes in the centre of these subunits seems far too high.

A potentially better method would be to rigid body fit the subunits and calculate the centre of mass of the c-ring and d-subunit. However, to make the comparison Figure (as in Figure 4), the states will need to be aligned so that all the c-ring axes are perpendicular to the plane of the figure. Otherwise, the differences between the centre of masses will change due to the viewing angle, rather than changes in the subunit positions.

We agree with your opinion. We have tested with the method you suggested below and updated the text with these measurements. After confirming a good rigid body fit of the d-subunit to the density, we used the UCSF Chimera “measure center” function for simulated maps of the c-ring and d-subunits, all generated onto the same map grid. Using the centre of mass of the c-ring and d-subunit, we calculated the X/Y offset relative to the central axis of the c-ring. While it deviates slightly from our original measurements it is not to a degree which we think affects our conclusions. And I agree, 0.1 Å was far too high a degree of accuracy.

From your recommendation, we calculated the offset using that method and have updated the text accordingly. Because of the shape of the d-subunit, the center of mass is slightly offset. The “measure center” variances were as follows:

Center of 2BL2 molmap: 174.23, 189.04 (Z dimension ignored)

State 1: X: 179.02, Y: 193.00	6.2 Å → 6 Å
State 1': X: 171.18, Y: 192.91	4.9 Å → 5 Å
State 2: X: 169.57, Y: 189.93	4.7 Å → 5 Å
State 2': X: 170.88, Y: 182.36	7.5 Å → 8 Å
State 3: X: 173.44, Y: 184.28	4.8 Å → 5 Å
State 3': X: 177.66, Y: 188.87	3.4 Å → 3 Å

The above numbers are directly taken from UCSF Chimera to 1 decimal places. We finally rounded to the nearest Angstrom.

These results have been replaced with red “+” in Fig. 4 and Fig. S6.

(iv) I requested the PDB validation reports for the maps, not for models. If the authors prefer, they could upload the raw maps with the submission?

My apologies, we misunderstood. The EMDB validation files named as EMD-xxx.pdf have been included. The maps should be public on EMDB shortly.

REVIEWERS' COMMENTS:

Reviewer #2 (Remarks to the Author):

The authors have addressed all my concerns, and I agree for its publication in Communications Biology.

Reviewer #3 (Remarks to the Author):

I recommend that this study be published.